# Comparative Analysis of Viral Load Quantification Using Reverse Transcription Polymerase Chain Reaction and Digital Droplet Polymerase Chain Reaction

**DOI:** 10.3390/ijms26020725

**Published:** 2025-01-16

**Authors:** Paula Gebe Abreu Cabral, Sávio Bastos de Souza, Raul Ferraz Arruda, Sheila Passos de Figueiredo Cabral, Arícia Leone Evangelista Monteiro de Assis, Yolanda Porto Muniz Martins, Antônio Brazil Viana Junior, Junbiao Chang, Pingsheng Lei, Renato Martins da Silva

**Affiliations:** 1High Complexity Center, Instituto Galzu, Campos dos Goytacazes 28110-000, RJ, Brazil; pgacabral99@gmail.com (P.G.A.C.); saviobas@gmail.com (S.B.d.S.); raul_arruda@hotmail.com (R.F.A.); sheila.figueiredo@icloud.com (S.P.d.F.C.); aricialeone@hotmail.com (A.L.E.M.d.A.); yolymartins@icloud.com (Y.P.M.M.); 2Complexo Hospital Universitário Walter Cantídio da UFC/Ebserh, Fortaleza 60430-372, CE, Brazil; brazil.estatistico@gmail.com; 3NMPA Key Laboratory for Research and Evaluation of Innovative Drug, Henan Normal University, Xinxiang 453007, China; changjunbiao@zzu.edu.cn; 4Institute of Material Medical, Chinese Academy of Medical Sciences & Peking Union Medical College, Beijing 100050, China; lei@imm.ac.cn

**Keywords:** COVID-19, SARS-CoV-2, AZVUDINE, FNC, viral load, DDPCR

## Abstract

In the year 2019, a highly virulent coronavirus named severe acute respiratory syndrome coronavirus 2 (SARS-CoV-2) emerged, precipitating the outbreak of the illness known as coronavirus disease 2019 (COVID-19). The commonly employed reverse transcription polymerase chain reaction (RT-qPCR) methodology serves to estimate the viral load in each patient’s sample by employing a standard curve. However, it is imperative to recognize that this technique exhibits limitations with respect to clinical diagnosis and therapeutic applications, since an advancement of the conventional polymerase chain reaction methods, digital polymerase chain reaction (digital PCR or DDPCR), enables the direct quantification and clonal amplification of nucleic acid strands. The primary divergence between dPCR and traditional PCR resides in their approaches to measuring nucleic acid quantities. In this study, we investigated the viral loads of severe acute respiratory syndrome coronavirus type 2 (SARS-CoV-2) within 461 participants. By employing both RT-qPCR and DDPCR techniques, we established a comparison between the quantification methodologies of the two approaches. Our findings illustrate that the quantification through DDPCR affords a superior means of monitoring viral load within lower respiratory tract samples, thus enhancing the assessment of disease progression, particularly in scenarios characterized by low viral loads.

## 1. Introduction

It is well established that severe acute respiratory syndrome coronavirus 2 (SARS-CoV-2) is the cause of the coronavirus disease 2019 (COVID-19) pandemic [1,2]. The extensive impact of COVID-19 on a global scale resulted in widespread fatalities and triggered the declaration of a worldwide public health emergency [3,4,5,6].

Reverse transcription polymerase chain reaction (RT-PCR) stands as the established benchmark for diagnosing the etiology of COVID-19 [7,8]. Yet, doubts emerged regarding the sensitivity and dependability of RT-PCR, given the instances of negative outcomes in patients strongly suspected to be afflicted, and positive outcomes in confirmed cases post-recovery [2,9]. Furthermore, the utility of RT-PCR for analyzing viral load in evaluating disease progression, prognosis, and the effectiveness of antiviral treatments is inherently limited.

Recently, a pair of randomized trials employing a double-blind design, running in parallel and controlled with placebos, were executed (Registration numbers for the trials: NCT04668235 and NCT05033145) [10,11], conducted between April 2021 and July 2022, with a total of 476 and 695 individuals, respectively [10,11]. The objective was to assess the effectiveness of an antiviral treatment against COVID-19. In order to forecast the clinical result of COVID-19, an analysis was conducted on the progression of viral load using RT-qPCR and DDPCR techniques. The present work makes use of data from these two previous studies.

RT-PCR (reverse transcription polymerase chain reaction) and ddPCR (droplet digital PCR) differ primarily in their methodologies and applications. RT-PCR is used to measure RNA levels by converting RNA into cDNA and amplifying it, with real-time quantification relying on fluorescence, making it suitable for gene expression analysis and viral RNA detection [7,8]. It requires a standard curve for absolute quantification and is sensitive to amplification efficiency. In contrast, ddPCR partitions the sample into thousands of droplets, performing PCR in each droplet for absolute quantification without standard curves. This method offers higher sensitivity and precision, making it ideal for detecting rare mutations and low-abundance targets, though it is more expensive and technically complex [9,12]. These features contribute to the increased precision and sensitivity of ddPCR compared to RT-qPCR [9].

Emerging evidence from multiple investigations suggests the superior attributes of droplet digital PCR (DDPCR) compared to RT-PCR, encompassing absolute quantification and heightened virus detection sensitivity [9,12]. This study encompassed a comparative analysis of RT-PCR and DDPCR for diagnosing COVID-19. Additionally, we scrutinized variations in viral load across diverse tissue samples, alongside their dynamics during the course of disease progression in SARS-CoV-2 patients.

## 2. Results

### Detection of SARS-CoV-2 Viral Load by RT-PCR and DDPCR Technique

A total of 453 sample patients (178 moderate and 275 mild cases of COVID-19) were tested by both RT-PCR and DDPCR, and the progression of disease was accompanied by both approaches, using nasal swabs.

During the course of the previous clinical studies, the antiviral-treated group showed no significant difference in viral load quantified through the RT-PCR technique compared to the placebo group. However, the viral load quantified by DDPCR shows a great difference between the groups (Table 1 and Table 2). The high sensitivity of the DDPCR confronts the variability obtained by calculating the viral load by RT-PCR (standard curve calculation due to the logarithmic variability) after treatment with AZVUDINE (FNC), showing a significant reduction in viral load in D3 (*p* < 0.002), D5, D7, and D9 (*p* < 0.001), and D11 (*p* < 0.006). Therefore, utilizing both RT-PCR and DDPCR techniques can effectively reflect the infection trend despite variations in values; in addition, correlation analyses were performed (as presented in Table 3 and Table 4, as well as Figure 1 and Figure 2).

A positive correlation exists in the quantification of viral load using the analyzed techniques, irrespective of treatment (Placebo or Azvudine) or disease severity (mild or moderate cases). This positive correlation is particularly pronounced between days D1 and D9, as indicated by coefficient of variation (RHO) values ranging from 0.65 to 0.88, all exhibiting *p*-values < 0.001 (Table 3 and Table 4). Nevertheless, it is worth noting that no correlation values were observed between days D11 and D13.

In Figure 1 and Figure 2 it is possible to observe the data dispersion between the viral loads quantified by the RT-PCR and DDPCR techniques. A positive correlation between the techniques is evident between days D1 and D9, irrespective of the use of Placebo or Azvudine. However, no correlation was observed between days D11 and D13.

## 3. Discussion

In numerous countries, the reverse transcription quantitative real-time polymerase chain reaction (RT-qPCR) technique has been employed for the detection and diagnosis of the virus [13]. Moreover, this method enables the estimation of viral load in samples obtained from individuals afflicted with this viral infection [13]. Illustratively, RT-qPCR has been utilized to report viral load through a standard curve in respiratory specimens, including throat swabs and nasopharyngeal swabs (NPSs), in patients affected by seasonal influenza or COVID-19 [10,11,12,14]. Nevertheless, doubts have been raised about the sensitivity and reliability of RT-PCR. Some patients with strong clinical symptoms and a history of exposure to the disease have received negative results, while certain confirmed cases have tested positive even after recovery [2,9].

The dataset from the previous clinical investigations has revealed a noteworthy congruence in the values derived from RT-PCR and DDPCR measurements across consecutive days for both studies (Table 1 and Table 2). However, a substantial disparity arises when comparing the viral load quantification results between RT-PCR and DDPCR for the same sample; the values obtained by DDPCR are unexpectedly much higher (Table 3 and Table 4, Figure 1 and Figure 2). Within the RT-PCR methodology, a standard curve was generated by employing serial dilutions of the positive control (SARS-CoV-2 viral RNA at a concentration of 1 × 10^4^ copies/μL) [10,11]. This discrepancy implies that the extent of infection in the patient surpasses initial estimations. This fact is attributed to the inherent nature of RT-PCR providing a logarithm to approximate the viral load calculation since the estimated viral load is constrained by this curve [15], preventing it from surpassing this limit, even if the virus’s quantity in a specific patient sample were greater, while DDPCR offers an actual count of viral copies (absolute quantification) [12,16,17].

This information alone substantiates the assertion that the assessment of qualitative viral load can effectively showcase the severity of each individual’s infection. Consequently, the heightened sensitivity of the DDPCR test illuminates the infection’s intensity, which can remain concealed within the results of RT-PCR.

This is an important aspect to consider, as it is not about replacing one method with another. Despite RT-PCR showing a lower sensitivity in estimating viral load through a standard curve, its application remains valuable for assessing the clinical progression of various viral infections [18]. Our goal is to establish a correlation since RT-PCR is much more widely used and accessible. The results suggest that this correlation is strong. If RT-PCR were used quantitatively, it would greatly aid in disease management. This becomes particularly relevant when a more precise approach, such as DDPCR, is not available.

In addition, a diminishing trend was observed in the viral load over successive days for both tests (Table 1 and Table 2). However, the reduction in viral load as observed through DDPCR is at a slower pace in the IGZ-1 study. The IGZ-1 study addressed COVID-19 moderate patients who required hospitalization, whereas the IGZ-2 study focused on COVID-19 mild cases. This discrepancy shows the potential for monitoring the severity and disease progression to effectively anticipate its worsening.

Altogether, the results of RT-PCR and DDPCR were consistent in the positive samples, and the viral load values of DDPCR and RT-PCR were highly correlated. Furthermore, the dynamic improved changes in the viral load during the course of COVID-19, as monitored by the DDPCR test, are more reliable and consistent compared to the RT-PCR test, with the aim of discerning strategies for the purpose of monitoring and mitigating the progression of mild and moderate instances of COVID-19.

## 4. Materials and Methods

### 4.1. Study Design

Prospective, randomized, double-blind, placebo-controlled clinical trials were performed at Santa Casa de Misericórdia of Campos dos Goytacazes Hospital, Sao Jose Hospital, Itaocara Hospital, Armando Vidal Hospital, Moacyr Gomes de Azevedo Hospital and Manoel Carola Hospital, with the first trial (NCT04668235) starting on 23 April 2021 and ending on 10 August 2022 with 476 participants [10], and the second trial (NCT05033145) starting on 15 January 2022 and ending on 27 July 2022 with 695 participants [11], totaling 1171 participants in both trials. Regarding patient recruitment, for trial NCT04668235, the first patient enrollment was on 23 April 2021, and the last patient enrollment was on 4 March 2022. For trial NCT05033145, the first patient enrollment was on 15 January 2022, and the last patient enrollment was on 22 March 2022. The strategic decision to centralize molecular biology analyses to ensure their standardization and quality was due to each RT-PCR equipment having its own sensitivity, and different kits of reagents for RT-PCR having different levels of performance. These trials was approved by the institutional review board of the National Health Surveillance Agency. Processes 25351.192286/2021-24 and 25351.319737/2021-87. The studies were approved by the National Council for Research Ethics, CAAE 39713120.9.0000.5244 and 52176421.8.0000.5244. The studies were also published in clinical trials (NCT04668235 and NCT05033145). All enrolled participants provided written informed consent, with the following eligibility criteria: age 18 and over, regardless of gender; respiratory or blood samples that tested positive for SARS-CoV-2 nucleic acid by RT-PCR, or respiratory or blood samples that tested highly homologous for the known SARS-CoV-2 by viral gene sequencing; the confirmation of COVID-19 according to the diagnostic criteria of “the latest Clinical guide-lines for novel coronavirus” issued by the World Health Organization (WHO) on 28 January 2020. Both studies have similar designs, with viral load analysis every 48 h, making it possible to compare their data.

Enrollment: After the application and signature of the ICF, total RNA was extracted using the MagMAX^TM^ Viral/Pathogen Nucleic Acid Isolation kit (Applied Biosystems, Waltham, MA, USA) according to the manufacturer’s instructions, using nasal and throat swabs from the clinical study participants. Then, the researcher evaluated whether the patient met the criteria, and enrolled the eligible patients.

This investigation encompassing a total of 453 moderate and mild COVID-19 participants. Nucleic acid detection analyses were performed every 48 h throughout the treatment period for optimal measurement of participant’s viral load. Two consecutive negative results configured clinical discharge.

### 4.2. Quantification of SARS-CoV-2 Viral Load by Reverse Transcription Polymerase Chain Reaction (RT-PCR)

Total RNA was extracted using the MagMAX^TM^ Viral/Pathogen Nucleic Acid Isolation kit (Applied Biosystems) according to the manufacturer’s instructions, using nasal and throat swabs from the clinical study participants.

Once total RNA was extracted, RT-PCRs were performed using the TaqPath^TM^ COVID-19 CE-IVD RT-PCR kit (ANVISA Reg.: 10358940107) according to the manufacturer’s instructions, using the QuantStudio5 RT-PCR equipment, Applied Biosystems, Berlin, Germany (ANVISA Reg: 10358940069). The primers and probes targeted the ORF1ab and N genes.

A standard curve was constructed using serial dilutions of the positive control (TaqPath^TM^ COVID-19 Control), which is SARS-CoV-2 viral RNA at a known concentration of 1 × 10^4^ copies/µL. The CTs obtained from each sample by RT-PCR were plotted on the standard curve to estimate the viral load of each sample.

A positive RT-PCR result occurs in CTs ≤ 30.5. In this case, when viral RNA is present, the specific probe used to detect SARS-CoV-2 is broken by DNA polymerase, emitting fluorescence. A high copy number of viral RNA generates high levels of fluorescence; therefore, the CT value appears earlier during the reaction. A low copy number of viral RNA generates low level of fluorescence, and consequently, the CT value appears later. Values of CTs > 30.5 are considered negative. By establishing a viral RNA concentration curve (present in the positive control), we will obtain a curve of CTs, from lower values (higher copies of viral RNA) to higher values (lower copies of viral RNA).

### 4.3. Quantification of SARS-CoV-2 Viral Load by Droplet Digital Polymerase Chain Reaction (DDPCR)

Total RNA was extracted using the MagMAX^TM^ Viral/Pathogen Nucleic Acid Isolation kit (Applied Biosystems) according to the manufacturer’s instructions, using nasal and throat swabs from the clinical study participants. Once total RNA was extracted, the DDPCR analyses were performed subsequently.

Reverse transcription PCR analyses were conducted with primers and probes targeting the ORF1ab and N genes and a positive reference gene. Reaction system and amplification conditions were performed according to the manufacturer’s specifications (Shanghai BioGem Medical Technology Co., Ltd., Shanghai, China).

Digital droplet PCR analyses were performed by the Targeting One Digital PCR System; COVID-19 digital PCR detection kit; droplet generation Kit; and Droplet detection kit. The kits allow the detection of the ORF1ab gene, N gene, and a positive reference gene. The detection limit was 10 copies/test. (Targeting OneTechnology (Beijing, China) is licensed by the China Food and Drug Administration.) A fractional number represents viral fragments that do not constitute a viral unit.

### 4.4. Statistical Analysis

The sample included 178 moderate and 275 mild COVID-19 participants. The mean value, standard deviation, quartiles, minimum, and maximum values for numerical variables were calculated. A correlation analysis of DDPCR viral loads in relation to RT-PCR was carried out using R Studio Software Version 3.6.0. Spearman’s Rank Correlation Coefficient (RHO) was used to assess the relationship between viral load measurements obtained by RT-PCR and ddPCR. Scatterplots were generated to visually represent the correlation between viral loads quantified by RT-PCR and ddPCR techniques for both mild and moderate cases of COVID-19. These plots illustrated the data dispersion and correlation between the two methods on different days.

## Figures and Tables

**Figure 1 ijms-26-00725-f001:**
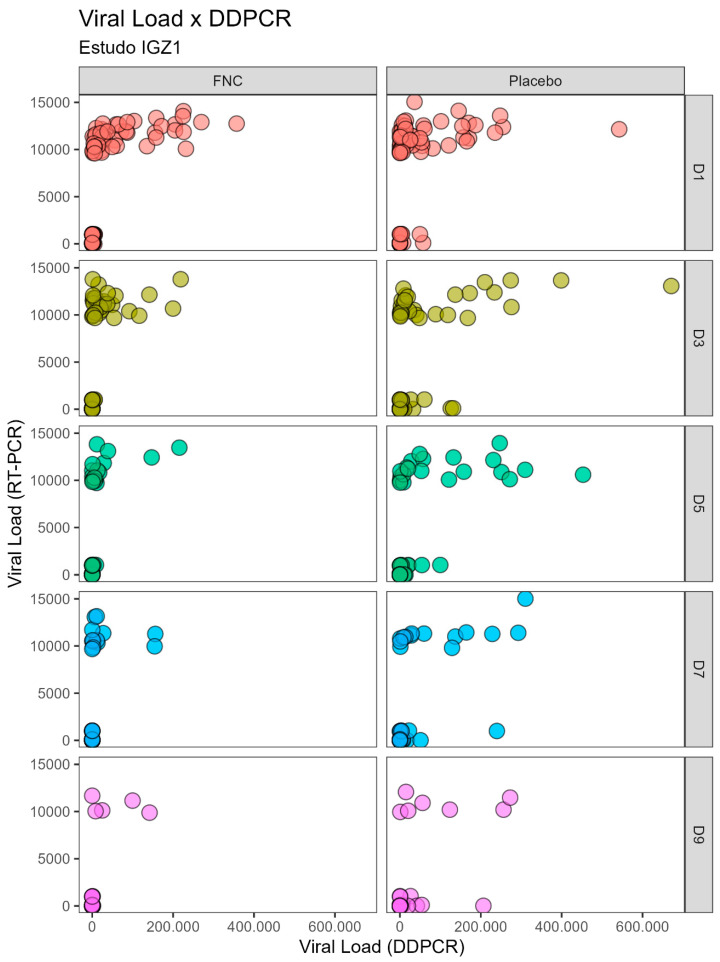
Scatterplots of correlation analysis between the viral load of SARS-CoV-2 quantified by the RT-PCR and DDPCR techniques for individuals with moderate disease severity (IGZ-1 Study) who received Azvudine or placebo on different days.

**Figure 2 ijms-26-00725-f002:**
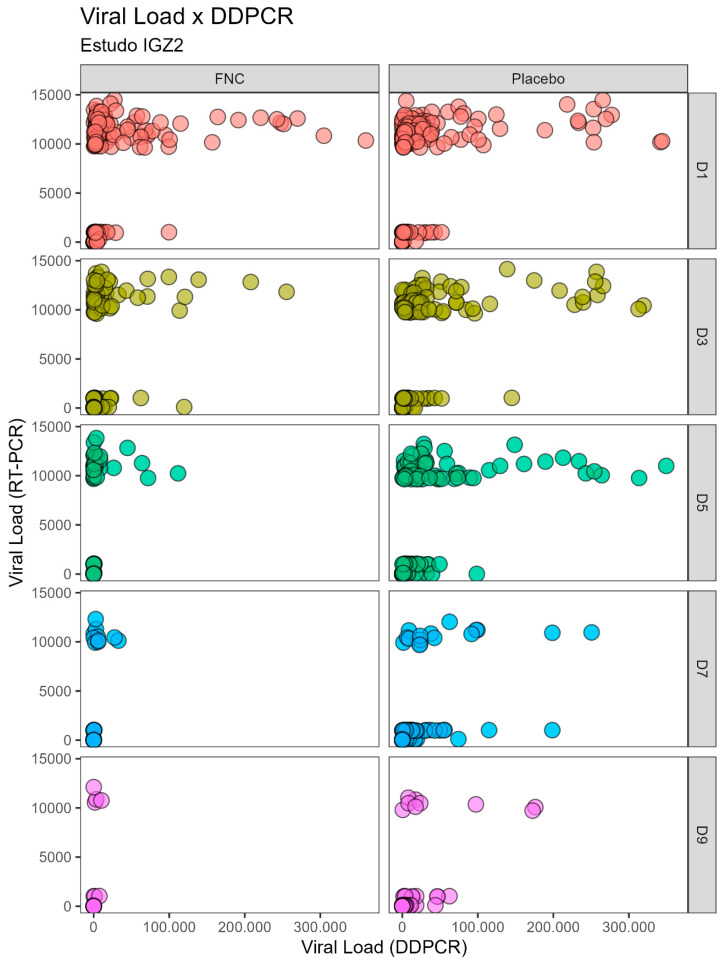
Scatterplots depicting correlation analysis between the viral load of SARS-CoV-2 quantified by the RT-PCR and DDPCR techniques for individuals with mild disease severity (IGZ-2 Study) who received Azvudine or placebo on different days.

**Table 1 ijms-26-00725-t001:** Average viral load values from IGZ-1 studies.

Variables	RT-PCR Total ^1^	DDPCR Total ^1^
Viral Load (D1)	7100 ± 5500 (10,200)	41,900 ± 78,600 (4900)
Viral Load (D3)	4100 ± 5300 (100)	26,900 ± 76,600 (180)
Viral Load (D5)	2700 ± 4700 (10)	20,900 ± 64,500 (20)
Viral Load (D7)	2500 ± 4500 (10)	21,900 ± 61,800 (10)
Viral Load (D9)	1400 ± 3500 (0)	15,700 ± 49,600 (0)
Viral Load (D11)	880 ± 2900 (0)	13,700 ± 40,700 (0)
Viral Load (D13)	480 ± 2100 (0)	7900 ± 26,400 (0)

^1^ Source: IGZ-1 published data (De Souza et al., 2023) [11].

**Table 2 ijms-26-00725-t002:** Average viral load values from IGZ-2 studies.

Variables	RT-PCR Total ^1^	DDPCR Total ^1^
Viral Load (D1)	8600 ± 4900 (10,800)	41,800 ± 74,800 (11,600)
Viral Load (D3)	5700 ± 5500 (1000)	38,700 ± 70,600 (11,500)
Viral Load (D5)	4300 ± 5200 (970)	38,800 ± 70,400 (9400)
Viral Load (D7)	1800 ± 3600 (60)	18,800 ± 41,300 (2500)
Viral Load (D9)	1300 ± 3300 (0)	12,400 ± 32,700 (880)
Viral Load (D11)	560 ± 2200 (0)	4400 ± 12,900 (70)
Viral Load (D13)	300 ± 1500 (0)	300 ± 1700 (0)

^1^ Source: IGZ-2 published data (Da Silva et al., 2023) [10].

**Table 3 ijms-26-00725-t003:** IGZ1 analysis of correlation between the viral load of the SARS-CoV-2 virus quantified by the RT-PCR X DDPCR technique for individuals with moderate disease severity (IGZ-1 Study), who were administered Azvudine or placebo on different days. RHO represents the correlation coefficient; 95% CI represents the 95% confidence interval, accompanied by the corresponding *p*-value.

Group	Day	RHO	95% CI	*p*
**Placebo**	D1	0.69	0.56–0.79	<0.001
	D3	0.84	0.76–0.90	<0.001
	D5	0.80	0.70–0.87	<0.001
	D7	0.74	0.58–0.85	<0.001
	D9	0.81	0.67–0.90	<0.001
**Azvudine**	D1	0.87	0.80–0.91	<0.001
	D3	0.88	0.82–0.92	<0.001
	D5	0.81	0.71–0.87	<0.001
	D7	0.74	0.60–0.84	<0.001
	D9	0.65	0.44–0.80	<0.001

**Table 4 ijms-26-00725-t004:** IGZ-2 analysis of correlation between the viral load of the SARS-CoV-2 virus quantified by the RT-PCR X DDPCR technique for individuals with mild disease severity (IGZ-2 Study), who were administered Azvudine or placebo on different days. RHO represents the correlation coefficient; 95% CI represents the 95% confidence interval, accompanied by the corresponding *p*-value.

Group	Day	RHO	95% CI	*p*
**Placebo**	D1	0.44	0.29–0.57	<0.001
	D3	0.63	0.51–0.72	<0.001
	D5	0.65	0.53–0.75	<0.001
	D7	0.81	0.72–0.87	<0.001
	D9	0.82	0.72–0.88	<0.001
**Azvudine**	D1	0.49	0.35–0.61	<0.001
	D3	0.77	0.69–0.83	<0.001
	D5	0.70	0.58–0.78	<0.001
	D7	0.87	0.79–0.92	<0.001
	D9	0.84	0.67–0.92	<0.001

## Data Availability

A preprint of this study has been deposited on Research Square [https://doi.org/10.21203/rs.3.rs-3380880/v1]. During the preparation of this work, no generative AI and AI-assisted technologies were used in the writing process.

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
