# Peer review of "Comparative Analysis of Viral Load Quantification Using Reverse Transcription Polymerase Chain Reaction and Digital Droplet Polymerase Chain Reaction"

_ijms, 2025, doi:10.3390/ijms26020725_

Round 1

Reviewer 1 Report

Comments and Suggestions for Authors

This manuscript outlines a study comparing two methodologies, reverse transcription-polymerase chain reaction (RT-PCR) and digital polymerase chain reaction (ddPCR), for quantifying the viral load of SARS-CoV-2 in lower respiratory tract samples.

The authors mentioned that patients were enrolled based on certain criteria, but it doesn't specify what those criteria were. Clear and specific inclusion and exclusion criteria are essential for ensuring the homogeneity of the study population and the validity of the results.

In Section 2.1, the throat swab specimens were collected for nucleic acid testing by RT-PCR confirmation. However in section 2.2, Total RNA was extracted using the MagMAXTM Viral/Pathogen Nucleic Acid Isola-tion kit (Applied Biosystems) according to the manufacturer's instructions, using nasal and throat swabs from the clinical study participants. Please clarify it.

Please replace figure 1 and 2 with high resolution ones.

Please make changes for figure 1 and 2 to indicate clearly which data is from RT-PCR or DDPCR.

major revision:

Line 103, Please explain why CTs ≤30,5 were considered as positive. According to WHO kit, we usually make 36 at detection limit. Please retest these samples with other high sensitivity kits.

Please indicate clearly the data source for table 1, 2, 3 and 4. They are original data or from references.

Comments on the Quality of English Language

English is fine

Author Response

REVIEWERS COMMENTS 1

Comments and Suggestions for Authors

This manuscript outlines a study comparing two methodologies, reverse transcription-polymerase chain reaction (RT-PCR) and digital polymerase chain reaction (ddPCR), for quantifying the viral load of SARS-CoV-2 in lower respiratory tract samples.

  1. The authors mentioned that patients were enrolled based on certain criteria, but it doesn't specify what those criteria were. Clear and specific inclusion and exclusion criteria are essential for ensuring the homogeneity of the study population and the validity of the results.

Answer: The Inclusion and exclusion criteria of patients, design, goals and outcomes are found in the supplementary methodology, as cited in the lines 95-96. However, now we also added in the methodology section in the lines 90-95.

  1. In Section 2.1, the throat swab specimens were collected for nucleic acid testing by RT-PCR confirmation. However in section 2.2, Total RNA was extracted using the MagMAXTM Viral/Pathogen Nucleic Acid Isola-tion kit (Applied Biosystems) according to the manufacturer's instructions, using nasal and throat swabs from the clinical study participants. Please clarify it.

Answer: We agree. We corrected the sentence (lines 99-102), total RNA was extracted using the MagMAXTM Viral/Pathogen Nucleic Acid Isola-tion kit, using nasal and throat swabs from the clinical study participants.

  1. Please replace figure 1 and 2 with high resolution ones. (Colocar figuras com maior qualidade de resolução)

Answer: The figures have been replaced and made available in higher resolution.

  1. Please make changes for figure 1 and 2 to indicate clearly which data is from RT-PCR or DDPCR.

Answer: Figures 1 and 2 correspond to the viral load data from the IGZ-1 and IGZ-2 studies, respectively. In these scatter plots representing correlation analysis, the RT-PCR data is plotted on the Y-axis, and the DDPCR data is plotted on the X-axis.

MAJOR REVISION:

  1. Line 103, Please explain why CTs ≤30,5 were considered as positive. According to WHO kit, we usually make 36 at detection limit. Please retest these samples with other high sensitivity kits.

Answer: High cycle threshold (Ct) values for severe acute respiratory syndrome coronavirus 2 (SARS-CoV-2) can indicate true infections or false-positive results. Numerous issues regarding the interpretation of high Ct values have been documented in the literature, highlighting the negative consequences of misinterpretation (Mitchell et al., 2022; Rondaan et al., 2023; Aranha et al., 2021).

The Brazilian Health Regulatory Agency (Anvisa), an autonomous entity under the Ministry of Health and part of the Brazilian National Health System (SUS), addressed this in a technical note from May 2020 titled "ANVISA – ACCURACY OF DIAGNOSTIC TESTS REGISTERED WITH ANVISA for COVID-19 for qualitative analysis." The note explains that each RT-PCR equipment has unique sensitivity and that different RT-PCR reagent kits perform variably. It suggests using lower Ct values to determine negative results.

Tests approved by the National Research Ethics Committees (CEP/CONEP) established a safe Ct value for negative results. Ct 30.5 was identified as the optimal threshold, and subsequent clinical studies on COVID-19 used this parameter (Da Silva et al., 2023; De Souza et al., 2023). These studies employed a QuantStudio5 RT-PCR system from Applied Biosystems (ANVISA Reg: 10358940069), using primers and probes targeting the ORF1ab and N genes. RT-PCR tests were conducted at 48-hour intervals. Clinical data from participants who reached a Ct value of 30.5 showed consistency. Additionally, absolute quantification by digital droplet PCR (DDPCR) in the same samples (with Ct 30,5) revealed a viral load of zero (Da Silva et al., 2023; De Souza et al., 2023).

References

Aranha C, Patel V., Bhor V., Gogoi D. Cycle threshold values in RT-PCR to determine dynamics of SARS-CoV-2 viral load: An approach to reduce the isolation period for COVID-19 patients. Journal of Medical Virology 2021. https://doi.org/10.1002/jmv.27206

Da Silva RM, Gebe Abreu Cabral P, de Souza SB, Arruda RF, Cabral SPF, de Assis ALEM, et al. Serial viral load analysis by DDPCR to evaluate FNC efficacy and safety in the treatment of mild cases of COVID-19. Front Med. (2023) 10:1143485. doi: 10.3389/fmed.2023.1143485

De Souza, B. S., Cabral, P. G. A., Da Silva, R. M., Ferraz, R. A., Cabral, S. P. F., Assis, A. L. E. M., Viana Junior, A. B., Degrave, W. M. S., Moreira, A. S., Silva, C. G., Chang, J., Lei, P. (2023). "Phase III, randomized, double-blind, placebo-controlled clinical study: a study on the safety and clinical efficacy of AZVUDINE in moderate COVID-19 patients." Frontiers in Medicine, 10. doi: 10.3389/fmed.2023.1215916. ISSN 2296-858X.

Mitchell SL, Loeffelholz MJ. Considerations regarding Interpretation of Positive SARS-CoV-2 Molecular Results with Late Cycle Threshold Values. J Clin Microbiol. 2022 Sep 21;60(9):e0050122. doi: 10.1128/jcm.00501-22. Epub 2022 Jun 6. PMID: 35658526; PMCID: PMC9491168.

Rondaan C, Gard L, Niesters HGM, van Leer-Buter C, Zhou X. COVID or no COVID: Interpreting inconclusive SARS-CoV-2 qPCR results in different populations and platforms. J Clin Virol Plus. 2023 Jun;3(2):100145. doi: 10.1016/j.jcvp.2023.100145. Epub 2023 Mar 9. PMID: 36941981; PMCID: PMC9997055.

Rao, S.N., Manissero, D., Steele, V.R. et al. A Systematic Review of the Clinical Utility of Cycle Threshold Values in the Context of COVID-19. Infect Dis Ther 9, 573–586 (2020). https://doi.org/10.1007/s40121-020-00324-3

  1. Please indicate clearly the data source for table 1, 2, 3 and 4. They are original data or from references.

Answer: The source of the data was a pair of randomized trials employing a double-blind design, running in parallel and controlled with placebos (Registration numbers for the trials: NCT04668235 and NCT05033145) [10,11], conducted between April 2021 and July 2022, with a total of 476 and 695 individuals, respectively. The resultsof both clinical studies had already been published (Da Silva et al., 2023; De Souza et al., 2023). A clearer information about the source was included in the introduction (lines 47-54).

.

Reviewer 2 Report

Comments and Suggestions for Authors

The manuscript compares viral load measurements measured using RT-PCR and ddPCR. There are some deficiencies in the manuscript that need to be addressed:

1. The introduction does not provide any background on the two tests. How are they different? Do we expect the measurements made by the two tests to be the same? What are some of the other benefits or disadvantages of using either test?

2. Acronyms are not capitalized in the section titles.

3. While the title and abstract suggest that the study is comparing viral load measurements made via two different tests, the data is presented as treated and untreated groups and it appears that those two groups are being compared. What is the actual purpose of this study?

4. The quality of figures 1 and 2 is rather poor. The images are so blurry that I cannot read the text.

5. What is meant by "The comparison of the two groups under general conditions was analyzed with appropriate methods according to the types of indicators"? The authors need to clearly specify what analysis was done so that someone else can reproduce their work.

6. The authors appear to have fit lines to the RT-PCR vs. ddPCR data, but the data does not look the least bit linear.

7. Line 165: RHO is not given in Tables 1 or 2.

8. How is RHO calculated?

9. The authors are inconsistent with how they are indicating decimals --- sometimes they use a point and sometimes a comma. In a few numbers in Tables 1 and 2 I believe the decimal point is missing entirely.

10. The uncertainty in Tables 1 and 2 should not have that many significant figures.

11. I'm assuming the authors calculated a linear correlation coefficient for this data. This is not appropriate since the data is clearly not linear.

12. The English is rather poor. There are issues with verb tense in some places and many sentences are structured badly.

Comments on the Quality of English Language

The manuscript needs editing by someone with better English language skills.

Author Response

REVIEWERS COMMENTS 2

The manuscript compares viral load measurements measured using RT-PCR and ddPCR. There are some deficiencies in the manuscript that need to be addressed:

  1. The introduction does not provide any background on the two tests. How are they different? Do we expect the measurements made by the two tests to be the same? What are some of the other benefits or disadvantages of using either test?

Answer: RT-PCR (Reverse Transcription Polymerase Chain Reaction) and ddPCR (Droplet Digital PCR) mainly differ in their techniques and uses. RT-PCR involves converting RNA into cDNA and amplifying it, with real-time fluorescent measurement, making it ideal for analyzing gene expression and detecting viral RNA. This method requires a standard curve for absolute quantification and is influenced by amplification efficiency. On the other hand, ddPCR divides the sample into numerous droplets, performing PCR in each droplet individually to achieve absolute quantification without the need for standard curves. This approach provides greater sensitivity and precision, suitable for detecting rare mutations and low-abundance targets, although it is more costly and complex.

A new paragraph about these differences was added in the introduction (Lines 55-64).

  1. Acronyms are not capitalized in the section titles.

Answer: done.

  1. While the title and abstract suggest that the study is comparing viral load measurements made via two different tests, the data is presented as treated and untreated groups and it appears that those two groups are being compared. What is the actual purpose of this study?

Answer: The primary purpose of this study is to compare the quantification of viral loads using RT-PCR and digital droplet PCR (ddPCR) in individuals infected with SARS-CoV-2. While the study includes both treated (with Azvudine) and untreated (placebo) groups, the main objective is to evaluate the performance and correlation between these two quantification methods across different conditions, including varying severities of the disease (mild and moderate cases).

  1. The quality of figures 1 and 2 is rather poor. The images are so blurry that I cannot read the text.

Answer: The figures have been replaced and made available in higher resolution.

  1. What is meant by "The comparison of the two groups under general conditions was analyzed with appropriate methods according to the types of indicators"? The authors need to clearly specify what analysis was done so that someone else can reproduce their work.

Answer: There was an error in the description of the statistical methodology. To clarify, the statistical analysis performed in our study included the calculation of mean values, standard deviations, quartiles, minimum, and maximum values for numerical variables. We conducted a correlation analysis of the DDPCR viral loads in relation to RT-PCR using R Studio Software. Specifically, we used Spearman’s Rank Correlation Coefficient (RHO) to assess the relationship between the viral load measurements obtained by RT-PCR and ddPCR.

Additionally, we generated scatterplots to visually represent the correlation between viral loads quantified by RT-PCR and ddPCR techniques for both mild and moderate cases of COVID-19. These plots illustrated the data dispersion and the strength of the correlation between the two methods on different days.

The methodology section has been updated to reflect these specific statistical analyses (Lines 147-155).

  1. The authors appear to have fit lines to the RT-PCR vs. ddPCR data, but the data does not look the least bit linear.

Answer: Thank you for your observation. The fitting of lines to the RT-PCR vs. ddPCR data in our scatterplots was not intended to imply a strict linear relationship. Instead, it was meant to illustrate the overall trend and correlation between the two quantification methods.

We used Spearman’s Rank Correlation Coefficient (RHO) to assess the relationship between the viral load measurements obtained by RT-PCR and ddPCR, which is a non-parametric measure and does not assume a linear relationship. This coefficient assesses the strength and direction of the monotonic association between the two variables.

The lines in the scatterplots serve as a visual aid to highlight the general trend observed between the two methods. We acknowledge that a non-linear model might also be appropriate to describe the relationship between RT-PCR and ddPCR data.

  1. Line 165: RHO is not given in Tables 1 or 2.

Answer: Thank you for your observation. There was an error in the referencing of the tables. The RHO values are actually presented in Tables 3 and 4, not in Tables 1 and 2. The manuscript has been revised to correct this reference.

  1. How is RHO calculated?

Answer: RHO, the Spearman’s Rank Correlation Coefficient, is calculated by ranking the data from both RT-PCR and ddPCR measurements, then assessing how well the ranks for the two variables correlate. Specifically, it measures the strength and direction of the monotonic relationship between the two sets of rankings. This calculation was performed using R Studio Software in our study.

  1. The authors are inconsistent with how they are indicating decimals --- sometimes they use a point and sometimes a comma. In a few numbers in Tables 1 and 2 I believe the decimal point is missing entirely.

Answer: We have reviewed and corrected the use of decimal points and commas to ensure consistency.

  1. The uncertainty in Tables 1 and 2 should not have that many significant figures.

Answer: We reviewed the uncertainties presented in Tables 1 and 2. We chose to use tables in order to present the results of our measurements with greater precision.

  1. I'm assuming the authors calculated a linear correlation coefficient for this data. This is not appropriate since the data is clearly not linear.

Answer: The correlation coefficient we calculated is the Spearman’s Rank Correlation Coefficient (RHO), not a linear correlation coefficient. Spearman’s RHO is a non-parametric measure that assesses the strength and direction of the monotonic relationship between two variables, which does not assume a linear relationship. This approach is appropriate for our data and accurately reflects the correlation between the RT-PCR and ddPCR measurements.

  1. The English is rather poor. There are issues with verb tense in some places and many sentences are structured badly.

Answer: An English revision of the manuscript was reviewed.

Reviewer 3 Report

Comments and Suggestions for Authors

In this submission, the authors compared two PCR methods in their ability to detect SARS-COV-2 in patient samples.  Their findings demonstrate that the ddPCR method is superior in detecting viral material in patient samples, research which is relevant after the recent global pandemic.  While the techniques are established and findings relevant to diagnostics, I have some minor comments and suggestions to improve the presentation and discussion.

Throughout text and in figures: lowercase the DD in DDPCR for consistency

The units for figure axes should be provided around the axis labels, the figures are also fairly low resolution and difficult to read   Quantitative units should be provided for the tables as well.  The tables do not need the preceding 0 (eg., replace Table 01 with Table 1, and the others)   In the discussion, it would be worthwhile to discuss some of the practical pros and cons of RT- vs. ddPCR in regards to cost, experimental background, equipment / reagents, clinical approvals / throughout, etc.

Author Response

REVIEWERS COMMENTS 3

In this submission, the authors compared two PCR methods in their ability to detect SARS-COV-2 in patient samples.  Their findings demonstrate that the ddPCR method is superior in detecting viral material in patient samples, research which is relevant after the recent global pandemic.  While the techniques are established and findings relevant to diagnostics, I have some minor comments and suggestions to improve the presentation and discussion.

  1. Throughout text and in figures: lowercase the DD in DDPCR for consistency

Answer: Capital DD was standardized throughout the manuscript.

  1. The units for figure axes should be provided around the axis labels, the figures are also fairly low resolution and difficult to read Quantitative units should be provided for the tables as well.  The tables do not need the preceding 0 (eg., replace Table 01 with Table 1, and the others)   In the discussion, it would be worthwhile to discuss some of the practical pros and cons of RT- vs. ddPCR in regards to cost, experimental background, equipment / reagents, clinical approvals / throughout, etc.

Answer: The figures have been replaced and made available in higher resolution. The zero (0) was removed of all tables. A new paragraph about the RT-PCR and DDPCR and pros and cons was added in the introduction.

Round 2

Reviewer 2 Report

Comments and Suggestions for Authors

Thank you to the authors for addressing many of my previous comments. I still have some concerns with the presentation of the results here.

1. SARS-CoV-2 is not capitalized properly in the section (subsection) titles.

2. There are still too many significant figures in the uncertainties --- there should be no more than one or two. The number of decimals presented in the numerical value of the parameter should match the number of decimal places in the uncertainty. 

3. I think the line needs to be removed from the graphs. The authors are not performing an assessment of the linear correlation of the measurements, so having the line there is somewhat misleading.

4. I don't think the authors can really claim that there is a correlation between the measurements. Even the most correlated data sets have a rho of 0.88 which is not a particularly strong correlation (not one that I would want to base medical decisions on) and some of the data sets have rather low correlation (<0.5).

5. The authors also state that the ddPCR measurement is the more accurate measurement. There is nothing presented in this manuscript that allows us to determine which of the two measurements is more accurate.

Comments on the Quality of English Language

Minor corrections are needed

Author Response

RESPONSE TO REVIEWER

Comments and Suggestions for Authors

Thank you to the authors for addressing many of my previous comments. I still have some concerns with the presentation of the results here.

  1. SARS-CoV-2 is not capitalized properly in the section (subsection) titles.

Answer: Thank you for your observation. The changes were done.

  1. There are still too many significant figures in the uncertainties --- there should be no more than one or two. The number of decimals presented in the numerical value of the parameter should match the number of decimal places in the uncertainty. 

Answer: We thank the reviewer for highlighting the problem with the significant numbers in the uncertainties. We reviewed the manuscript and identified an error in the use of ".", so we replaced it with "," since numbers identify thousands, not decimal numbers.

  1. I think the line needs to be removed from the graphs. The authors are not performing an assessment of the linear correlation of the measurements, so having the line there is somewhat misleading.

Answer: Thank you for your suggestion. The suggested changes were made.

  1. I don't think the authors can really claim that there is a correlation between the measurements. Even the most correlated data sets have a rho of 0.88 which is not a particularly strong correlation (not one that I would want to base medical decisions on) and some of the data sets have rather low correlation (<0.5).

Answer: We appreciate the comments on the correlation values. However, according to Gignac and Szodorai (2016), it is essential to understand that many external variables can systematically reduce correlation coefficients, especially in studies that involve complex and multifactorial characteristics such as those examined in our research.

The interpretation of correlation sizes often follows traditional benchmarks, such as those proposed by Cohen (1988), where a correlation of 0.10 is considered small, 0.30 medium and 0.50 large. However, these guidelines have been re-evaluated in light of the results of empirical research, particularly in studies involving complex biological systems.

Gignac and Szodorai (2016) make a compelling argument that traditional effect size thresholds are overly conservative, especially in the context of investigating individual differences. Their empirical analysis suggests adjusting these thresholds to lower values, to 0.10, 0.20 and 0.30, for small, typical and relatively large correlations, respectively. This adjustment reflects the complexities of the real world, where obtaining very high correlations is uncommon due to the inherent variability in human behavior and external influences on the data.

Thus, our results, with a correlation of rho 0.88, exceed these reference values, indicating a relatively high association between the measured variables. It can be considered relatively high, as external factors and individual differences normally reduce observable correlation coefficients.

References:

Gignac, G. E., & Szodorai, E. T. (2016). Effect size guidelines for individual differences researchers. Personality and Individual Differences, 102, 74-78. https://doi.org/10.1016/j.paid.2016.06.069

  1. The authors also state that the ddPCR measurement is the more accurate measurement. There is nothing presented in this manuscript that allows us to determine which of the two measurements is more accurate.

Answer: Thank you for your comment. We acknowledge that previous studies have demonstrated the superior precision and sensitivity of ddPCR compared to RT-qPCR (Hayden et al. 2013; Yu et al., 2020). However, these studies were often limited in sample size and did not specifically address the impact of lower and higher viral load on assay performance. Our study leverages a comprehensive dataset obtained from two clinical studies (Da Silva et al., 2023; De Souza et al., 2023) to demonstrate that ddPCR quantification offers a superior method for monitoring viral load in lower respiratory tract samples, particularly in cases with low viral loads. This enhanced monitoring capability can significantly improve the assessment of disease progression. The partitioning nature of ddPCR, which divides the sample into thousands of individual droplets, enables absolute quantification of the target, eliminating the need for standard curves and mitigating the impact of inhibitors. These features contribute to the increased precision and sensitivity of ddPCR compared to RT-qPCR.

Previous research has also suggested the superiority of ddPCR, especially in quantifying low abundance targets and detecting rare allelic variations (Hindson et al. 2011; Pinheiro et al. 2012; Hayden et al. 2013). The technique has proven to be more robust against sample variability and inhibitors, factors that can compromise the accuracy of RT-qPCR. Our study further investigates the relationship between RT-PCR and ddPCR in the context of SARS-CoV-2 infection in patients with high and low viral loads. We demonstrate that ddPCR provides superior disease management, with a high correlation between ddPCR and RT-PCR viral load values. Importantly, we show that the dynamic changes in viral load monitored by ddPCR are more reliable and consistent than RT-PCR, aiding in the development of strategies for monitoring and mitigating the progression of mild and moderate COVID-19 cases.

We hope this expanded explanation clarifies our rationale for utilizing ddPCR and highlights the significant contributions of our study to the field.

References

Hindson BJ, Ness KD, Masquelier DA, Belgrader P, Heredia NJ, Makarewicz AJ, et al. High-throughput droplet digital PCR system for absolute quantitation of DNA copy number. Anal Chem. (2011) 83:8604–10. doi: 10.1021/ac202028g

Pinheiro LB, Coleman VA, Hindson CM, Herrmann J, Hindson BJ, Bhat S, et al. Evaluation of a Droplet Digital Polymerase Chain Reaction Format for DNA Copy Number Quantification. Anal Chem. (2012) 84:1003–11. doi: 10.1021/ac202578x

Hayden RT, Gu Z, Ingersoll J, Abdul-Ali D, Shi L, Pounds S, et al. Comparison of droplet digital PCR to real-time PCR for quantitative detection of Clostridium difficile. J Clin Microbiol. (2013) 51:530–6. doi: 10.1128/JCM.02565-12

Yu F, Yan L, Wang N, Yang S, Wang L, Tang Y, et al. Quantitative detection and viral load analysis of SARS-CoV-2 in infected patients. Clin Infect Dis. (2020) 71:793–8.doi: 10.1093/cid/ciaa345

Round 3

Reviewer 2 Report

Comments and Suggestions for Authors

The uncertainties still have too many significant figures. For example the first value in Table 2 (currently presented as 8,631 ± 4,897) should be 8,600 ± 4,900 if using two significant figures in the uncertainty.

I still don't believe that the authors can claim correlation between the measurements. The article cited by the authors in their response was asking the question, "at what r values can we be certain that the value of our dependent variable is partially determined by our independent variable?" This is a much lower standard than the what the authors are trying to get at in this manuscript. In this manuscript, the authors are comparing diagnostic medical tests and asking whether they are giving the same information about the patient's illness. They are really trying to determine if these tests have the same predictive power --- that is a much higher standard. The FDA would not approve a diagnostic test where the disease state was only 30% correlated to the outcome of the test.

On the final point about the accuracy of the ddPCR, thank you for the explanation. Please include some of that in the manuscript when you comment on the accuracy of the ddPCR test.

Author Response

RESPONSE TO REVIEWER

Comments and Suggestions for Authors

  1. The uncertainties still have too many significant figures. For example the first value in Table 2 (currently presented as 8,631 ± 4,897) should be 8,600 ± 4,900 if using two significant figures in the uncertainty.

Answer: Since the results are in the thousands, we initially preferred to include the complete numbers as they were extracted from the machine. However, following the reviewer's suggestion, we have rounded the amounts. The tables in the article have been revised accordingly.

  1. I still don't believe that the authors can claim correlation between the measurements. The article cited by the authors in their response was asking the question, "at what r values can we be certain that the value of our dependent variable is partially determined by our independent variable?" This is a much lower standard than the what the authors are trying to get at in this manuscript. In this manuscript, the authors are comparing diagnostic medical tests and asking whether they are giving the same information about the patient's illness. They are really trying to determine if these tests have the same predictive power --- that is a much higher standard. The FDA would not approve a diagnostic test where the disease state was only 30% correlated to the outcome of the test.

Answer: Thank you for your comments. We appreciate your concern regarding the correlation between the measurements and the standards we are aiming for in our manuscript.

This is indeed an important aspect to consider, as our goal is not to replace one method with another. While RT-PCR demonstrates lower sensitivity in estimating viral load through a standard curve, it remains a valuable tool for assessing the clinical progression of various viral infections. Given that RT-PCR is more widely used and accessible, establishing a correlation is crucial. Our findings shows that the correlation is not very strong, but it is relatively strong, and given the positive relationship, RT-PCR can be used as an alternative for clinical management of the disease when ddPCR is not available.

We understand your point about the higher standard required when comparing diagnostic medical tests for their predictive power. Our aim is to demonstrate that, despite the limitations of RT-PCR, it can still provide meaningful clinical insights, especially in situations where more precise methods, such as DDPCR, are not available.

We have revised the manuscript to better reflect this perspective and to clarify the strength and implications of the correlation we are reporting, and a new paragraph was included in the discussion (lines 306-312).

  1. On the final point about the accuracy of the ddPCR, thank you for the explanation. Please include some of that in the manuscript when you comment on the accuracy of the ddPCR test.

Answer: Thank you again. Additional information regarding the accuracy of the ddPCR test and its comparison with RT-PCR has been included in the Introduction section (Lines 55-65).
